# BendVLM: Test-Time Debiasing of Vision-Language Embeddings

**Walter Gerych**[1]     **Haoran Zhang**[1]     **Kimia Hamidieh**[1]     **Eileen Pan**[1]

**Maanas Sharma**[1]     **Thomas Hartvigsen**[2]     **Marzyeh Ghassemi**[1]

[1]MIT, [2]University of Virginia
`{wgerych, haoranz, hamidieh, eileenp, maanas, mghassem}@mit.edu,`
`hartvigsen@virginia.edu`

## Abstract

Vision-language model (VLM) embeddings have been shown to encode biases present in their training data, such as societal biases that prescribe negative characteristics to members of various racial and gender identities. VLMs are being quickly adopted for a variety of tasks ranging from few-shot classification to text-guided image generation, making debiasing VLM embeddings crucial. Debiasing approaches that fine-tune the VLM often suffer from catastrophic forgetting. On the other hand, fine-tuning-free methods typically utilize a "one-size-fits-all" approach that assumes that correlation with the spurious attribute can be explained using a single linear direction across all possible inputs. In this work, we propose BEND-VLM, a nonlinear, fine-tuning-free approach for VLM embedding debiasing that tailors the debiasing operation to each unique input. This allows for a more flexible debiasing approach. Additionally, we do not require knowledge of the set of inputs *a priori* to inference time, making our method more appropriate for online, open-set tasks such as retrieval and text guided image generation.[1]

## 1 Introduction

**Background.** Pretrained foundation Vision-language models (VLMs) such as CLIP [33], BLIP [22], and LLaVA [25] have seen wide adoption for tasks like image retrieval [21], zero and few-shot classification [33, 4], text-guided image generation [32], and facial recognition [58]. But VL models also encode societal biases [5, 27, 43, 49, 53]. As more and more systems rely on CLIP, the encoded representational harm [12, 3, 15, 52] can lead to allocative harm [34, 46, 14, 51, 16, 29], such as `Black` individuals being three times more likely to be misclassified into a nonhuman category by computer vision systems [1].

**State of the art.** Debiasing VLMs is an active area of research [6, 10, 20, 19, 50, 28]. One common approach is finetuning the embedding models to remove spurious correlations [59, 2, 42]. However, finetuning often decreases accuracy and generalizability of foundation models [31]—a significant drawback as these models are commonly used for zero-shot tasks. Most existing finetuning-free methods learn debiasing transformations of the initial text embeddings, but typically use one-size-fits-all *linear* debiasing functions that apply the same fixed transformation to every input [6, 10, 50].

While recent work has explored nonlinear VLMs [11], their method assumes access to the set of classes at test-time, requiring the debiasing training pipeline to be rerun if a query for a new class is

---

[1]code: https://github.com/waltergerych/bend_vlm

made. This is a major limitation in practice because many tasks VLMs are used for are often naturally *open-set*, where the classes to be evaluated for at test-time are unknown prior to inference.

**Problem Definition.** We study online, open-set debiasing for VLM embeddings. In this setup, we only have access to a VLM, along with a single-modal image dataset. This image dataset is only for the purpose of "training", and is not the dataset that the downstream task will work on. We assume that this dataset, which we call the *reference* dataset, has labels for the protected attribute(s) of interest. During test-time, we receive online input queries one at a time. These queries are also open-set, meaning that the classes or concepts they refer to are not known to us beforehand. For instance, the query may be "a photo of a nurse", but we do not have knowledge that nurse is a potential class of interest before receiving the query. Our goal is to debias the query embedding from the VLM in such as way that it does not more strongly associate the query embedding with any protected attribute value over another. For instance, the embedding for "a photo of a nurse" should not be more associated with images of women than with men.

**Challenges.** Online, open-set VLM debiasing is a challenging task. First, we must overcome *catastrophic forgetting*—a solution that debiases the embeddings, but degrades performance. Second, the interaction between protected attributes and query classes may be *nonlinear and instance-dependent*. For example, the transformation required to remove the gender bias from the embedding of "nurse" is likely not the same as the one to untangle gender bias associated with the embedding of "handyman". Third, queries from *open-set classes* means that our approach must be flexible enough to remove the association of protected attributes from classes unknown prior to inference time. Lastly, *online* settings demand computational efficiency and thus rule out refitting the debiasing component for each now class or query.

**Proposed approach.** We propose **B**ias **E**limination with **N**onlinear **D**ebiasing of **V**ision **L**anguage **M**odels (BEND-VLM), a test-time VLM debiasing method that leaves the VLM's weights unchanged, being efficient enough for online streaming queries. By using the easy-to-get pre-debiasing reference dataset with protected attributes, BEND-VLM allows for unsupervised test-time debiasing. On a high level, BEND-VLM consists of two main parts:

First, given an online query, we generate augmented queries that introduce protected attribute information. For example, given "a photo of a nurse" we generate "a photo of a {ATTRIBUTE} nurse", filling in {ATTRIBUTE} with male / female / nonbinary for gender debiasing, for instance. We get these augmented queries from a small language model, and use them to find the directions in the embedding space for that specific query that are most associated with the protected attribute. Given these directions, we project the embedding such that it is orthogonal to the protected attribute dimension, resulting in the first-stage debiased representation.

For the second step, we make use of the reference image dataset. We find the images in this dataset that are most associated with the query, and then subset them by protected attribute value. We find an updated, debiased query representation by solving a constrained optimization equation with the goal of finding an embedding with minimal distance to the first-stage debiased representation while being equally similar to the example images for each attribute value. For instance, we find an embedding that is equally similar to the nearest images for each gender. The resulting embedding will have little to no excess association with any of the debiased protected attribute values over any other. The output can then be passed to the downstream task.

**Contributions.**

- We introduce BEND-VLM, a novel test-time VLM debiasing approach that does not require finetuning.
- We propose a technique for finding local attribute subspaces specific to each query on-the-fly.
- We introduce a novel method for equalization by using a reference image dataset.
- We experimentally evaluate for classification, retrieval, and image captioning settings, showing BEND-VLM consistently outperforms the compared approaches.

## 2   Problem Definition

Let $(\mathbf{m}, \mathbf{t}, \mathbf{c}, \mathbf{a})$ be an (*image, text, class, attribute*) tuple distributed according to $P_M \times P_T \times P_C \times P_A$, a joint distribution over images, texts, classes, and attributes. Using the running example of nurses,

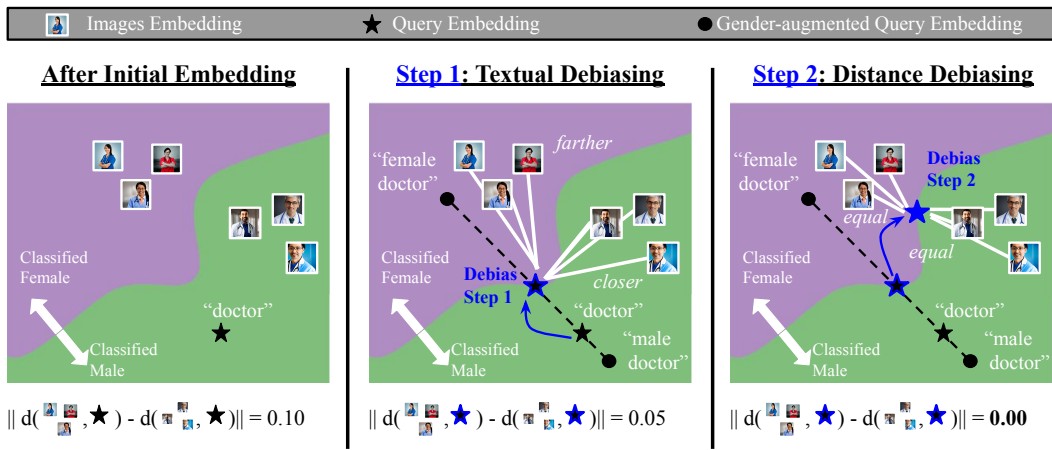

Figure 1: Overview of our two-step BEND-VLM method. In this example, the initial query embedding of doctor is more strongly associated with males, and the CCF distance is $0.10$. After performing debiasing step 1, Orthogonalizing the Embedding, the embedding is modified to remove bias along the gender direction defined by "male doctor" and "female doctor". This still results in a CCF distance of $0.05$. We then perform the second debiasing step, where the query embedding is again modified to be equidistant to the relevant male and female images. The final representation achieves the optimal distance of **0.00**.

a realization of $m$ could be an image of a nurse, $t$ the text "a photo of a nurse", $c$ the class nurse, and $a$ a protected attribute such as gender. Importantly, we do not assume that $\mathbb{C}$, the support of $P_C$, is known. This means we do not know what classes the user will query for during inference, and do not have access to a training set with these class labels.

Let $f_\theta^T : \mathbb{T} \to \mathbb{R}^d$ represent the text embedding model (e.g., CLIP's image encoder) and $f_\theta^M : \mathbb{M} \to \mathbb{R}^d$ represent the image encoder, where $\mathbb{T}$ and $\mathbb{M}$ are the text and image domain, respectively. We will use $f_\theta = \{f_\theta^T, f_\theta^M\}$ when referring to the VL model in general, rather than its modality-specific encoders. $f_\theta$ is used to obtain $d\big(f_\theta^M(m), f_\theta^T(t)\big)$, where $d(\cdot, \cdot)$ is a distance metric such as *cosine distance*. In practice, these (*image*, *text*) distance scores are used for zero-shot classification or image retrieval.

Let $t_c \in \mathbb{T}$ be a textual instance relating to class $c$. For instance, class $c$ could be nurse and $t_c$ "a picture of a nurse". Then, **our goal is to obtain a text embedding $z_c^* \in \mathbb{R}^d$ that is *Class Conditionally Fair***.

**Definition 1** (Class Conditionally Fair (CCF)). *A text embedding $z_c^*$ is **Class Conditionally Fair** for embedding model $f_\theta$, class $c$, and metric $d$ if for all $a_i, a_j \in \mathcal{A}$ the following holds:*

$$\mathbb{E}_{\mathbf{m}|a_i, c}\big[d(f_\theta^M(\mathbf{m}'), z_c^*)\big] = \mathbb{E}_{\mathbf{m}'|a_j, c}\big[d(f_\theta^M(\mathbf{m}'), z_c^*)\big].$$

Intuitively, a text embedding is CCF for class $c$ if the expected similarity between the text representation and *relevant* image embeddings — image embeddings that are also associated with class $c$ — is independent of the protected attribute value $a$. For instance, an embedding of the query "a picture of a nurse" is CCF if its expected similarity score for pictures of *female* nurses is equal to the expected similarity score for *male* nurses.

We also define **Class Conditionally Fair Distance** as a measure from how far off an embedding is from being CCF:

**Definition 2** (Class Conditionally Fair Distance). *The Class Conditionally Fair Distance for a text embedding $z_c$ class $c$, and metric $d$ is given by:*

$$d_{CCF}(z_c, c) = ||\mathbb{E}_{\mathbf{m}|a_i, c}\big[d(f_\theta^M(\mathbf{m}'), z_c)\big] - \mathbb{E}_{\mathbf{m}'|a_j, c}\big[d(f_\theta^M(\mathbf{m}'), z_c)\big]||_1.$$

The CCF distance of $z_c$ is 0 if and only if $z_c$ is CCF. In practice, we can't exactly compute the expectations in the CCF distance definition. Instead, these expectations can be replaced with the average distances from relevant samples in the evaluation dataset.

**Reference and Target Datasets.** In practice, we assume that we have a dataset $D_{\text{ref}} = \{(\boldsymbol{m}_i, \boldsymbol{a}_i)\}_{i=1}^N$ consisting of $N$ images with labeled attributes. For instance, $D_{\text{ref}}$ could be a dataset of pictures of people with corresponding `gender`, `race`, or `age` labels[2]. We focus on both the *image retrieval* and *zero-shot classification* setting. This *reference* dataset will be used to obtain debiased text embedding, as we describe in detail in the following section. We refer to the downstream dataset to be used in retrieval or zero-shot applications as the *target* dataset $D_{\text{target}} = \{\boldsymbol{m}_j\}_{j=1}^{N_{target}}$. $D_{\text{target}}$ is not available prior to inference.

**For retrieval**, we assume that $D_{\text{target}}$ is an unlabeled dataset of images, such that we want to retrieve images from this dataset that relate to streaming, open-set queries. For instance, the queries can be free-form text searches coming from a search engine user. In this open-set scenario the set of classes $\mathbb{C}$ is unknown — we do not know what classes users will search for *a priori*.

**For zero-shot classification**, we likewise focus on the streaming, open-set scenario. Images from $D_{\text{target}}$ will be compared against a set of texts $\{\boldsymbol{t}_{c0}, \boldsymbol{t}_{c1}, \cdots, \boldsymbol{t}_{cK}\}$ for the purpose of classification, where this set of texts relates to classes $\boldsymbol{c}_1, \boldsymbol{c}_2, \ldots, \boldsymbol{c}_K \in \mathbb{C}$, where $\mathbb{C}$ is unknown to us and potentially variable. For instance, a user may first wish to obtain zero-shot predictions of `hair color` of the portraits in $D_{\text{target}}$, and later wish to obtain predictions of whether the individuals have `eyeglasses`.

In both settings, we make the simplifying assumption that each user query $\boldsymbol{t}_c$ does not explicitly reference the protected attribute of interest. For instance, the query is `"a picture of a nurse"`, not `"a picture of a male nurse"` — and thus it is desirable for the query embedding to *not* be more associated with a particular gender. In the case where the query *does* contain explicit reference to $\boldsymbol{a}$ — `"a picture of a male nurse"` — it is straightforward to abstain from debiasing by using a language model to filter out these queries, or by checking for explicit attribute terms [3].

## 3 Methodology

On a high level, our BEND-VLM approach consists of a two-phase debiasing pipeline. We perform an initial debiasing pass by first employing the classic approach of orthogonalizing $f_\theta(\boldsymbol{t})$ to the attribute subspace $\boldsymbol{v}$ [24, 9]. However, unlike most prior works, we do *not* assume that the attribute subspace is globally constant for all queries; it may be the case that the direction in the embedding space corresponding to `gender` that differentiates `"a picture of a male nurse"` from `"a picture of a female nurse"` may not be equivalent to the gender direction between `"a picture of a baby boy"` and `"a picture of a baby girl"`. We find these *local attribute subspaces* using our ATTRIBUTEAUGMENT module to obtain attribute augmented versions of $\boldsymbol{t}$. After this first phase, we are left with the partially-debiased embedding $\boldsymbol{z}_c'$.

Our second and final debiasing pass consists of equalizing the distances between the embedding and relevant images from the reference dataset $D_{ref}$ belonging to each attribute class. We obtain the final debiased embedding $\boldsymbol{z}_c^*$ through an analytical solution to a constrained optimization equation.

### 3.1 Step 1: Making The Embedding Orthogonal To Local Attribute Subspace

Orthogonalizing text embeddings with respect to an attribute subspace, such as setting embedding dimensions corresponding to `gender` or `race` equal to zero, is a classic approach used for standard text embeddings [24, 9] and has recently shown promise in debiasing VL models [10]. Whereas existing approaches typically find a single attribute subspace for instances, we find local attribute subspaces in addition to the global subspace.

Let $\boldsymbol{t}_c$ be the initial text query coming in to the system. We then obtain $\boldsymbol{t}_{c,a_i}$ for all $\boldsymbol{a}_i \in \mathcal{A}$. For instance, if $\boldsymbol{a}$ refers to gender and $\boldsymbol{t}_c = $ `"a picture of a nurse"`, then we would obtain `"a picture of a male nurse"` and `"a picture of a female nurse"` for $\boldsymbol{t}_{c,a_{male}}$ and

---

[2]In a practical application, these protected attributes could be noisy labels assigned by an attribute predictor. For instance, gender labels could be obtained by using CLIP for zero-shot gender prediction.

[3]e.g. we could filter for `gender` with `GenderspacY`: https://github.com/sidatasciencelab/gender-spacy

$t_{c,a_{female}}$, respectively. We draw each $t_{c,a_i}$ from our ATTRIBUTEAUGMENT module: $\{t_{c,a_i}\}_{i\in\mathbb{A}} = $ ATTRIBUTEAUGMENT$(t_{c,a_i};\mathbb{A})$. In practice, we use an LLM to instantiate ATTRIBUTEAUGMENT. In a lower resource setting, ATTRIBUTEAUGMENT could feasibly be implemented through simpler text processing techniques to identify the subject of the query and insert corresponding attribute strings before the subject; e.g. inserting `"male"` and `"female"` before the subject for gender debiasing.

Let $A$ be a matrix whose columns are $f_\theta^T(t_{c,a_i}) - f_\theta^T(t_c)$ for $i = 1 \to |\mathbb{A}|$. To combat potential noise from estimating the local attribute subspace, we additonally include generic attribute text embeddings into the columns of $A$ as well. For instance, for gender debiasing we include the embeddings of `"a picture of a man"` and `"a picture of a woman"`. We then obtain the initial debiased embedding $z_c'$ as:

$$z_c' = V f_\theta^T(t_c),$$

where $V = I - A(A^\top A)^{-1} A^\top$ is the orthogonal projection matrix of $A$ [10].

Importantly, despite $z_c'$ being orthogonal to the local attribute subspace it is *not* necessarily equally similar to the image embeddings of relevant instances when conditioned on the "debiased" attribute.

---

**Lemma 1** (Orthogonalization does not yield Class Conditional Fairness.)**.** *The following does not hold in general:*

$$\mathbb{E}_{\mathbf{m}|a_i,c}\big[d(f_\theta^M(\mathbf{m}'), z_c')\big] = \mathbb{E}_{\mathbf{m}'|a_j,c}\big[d(f_\theta^M(\mathbf{m}'), z_c')\big].$$

---

We show an example of this in Figure 1, where we see that step 1 does not result in significantly improved CCF distances. To mitigate this, we propose a second debiasing step.

## 3.2  Step 2: Using Reference Images To Equalizing the Text Embedding

In this second stage, we equalize the distances between the images in $D_{\text{ref}}$ and the debiased embedding $z_c'$, with the goal of making relevant images from each attribute group equally similar to the text embedding. Let $D_{\text{ref}}(a_i, c)$ be images in the reference dataset that are associated with attribute class $a_i$ and class $c$. We want to find the embedding $z_c^*$ that satisfies the following set of conditions $\mathcal{C}$:

$$\mathcal{C} = \left\{ \frac{\sum_{m_j \in D_{\text{ref}}(a_i,c)} d(f_\theta^M(m_j, z_c^*))}{|D_{\text{ref}}(a_i,c)|} = \frac{\sum_{m_k \in D_{\text{ref}}(a_1,c)} d(f_\theta^M(m_k, z_c^*))}{|D_{\text{ref}}(a_1,c)|} \right\}_{i=1\to|\mathbb{A}|}$$

These constraints say that the average distance between relevant image embeddings should be equal for all attribute value splits. For example, the distance between the embedding of `"a picture of a nurse"` and relevant `male` images should match the distance between the embedding and relevant `female` images.

Note that since we do not assume access to context labels for $D_{\text{ref}}$, it is not immediately obvious on how to obtain each $D_{\text{ref}}(a_i, c)$. Instead, $D_{\text{ref}}(a_i, c)$ is by selecting $n$ images with attribute value $a_i$ that are most similar to the query embedding $z_c'$. The value of $n$ could be found using change-point detection, such that $n$ is the value where the elbow in the plot of similarity over indexes sorted by similarity score [38]. A less sophisticated approach — but one we find works well in practice — is to simple chose $n$ as a hyperparameter, and use the same value for each attribute and query.

Finding any embedding that satisfies $\mathcal{C}$ is not enough, since we want to ensure that the debiased embedding does not lose information unrelated to the protected attribute $a$. This means we want to find a debiased embedding with minimal distance to the previous embedding. We want to find a $z_c^*$ that minimizes distance to the first-pass debiased $z_c'$:

$$\mathcal{L}_{\text{initial}} = d(z_c^*, z_c')$$

We thus find $z_c^*$ by solving the following constrained optimization equation:

$$z_c^* = \arg\min_{z_c^*} \mathcal{L}_{\text{initial}}, \text{ under the set of constraints } \mathcal{C}. \tag{1}$$

Equation 1 has a simple analytical solution for the binary attribute case, when $d(\cdot, \cdot)$ is cosine distance and each embedding has unit norm length.

**Lemma 2.** *The value of $\boldsymbol{z}_c^*$ that minimizes the distance from the initial embedding $\boldsymbol{z}_c'$ while satisfying the image-embedding fairness constraint is:*

$$\boldsymbol{z}_c^* = \frac{\boldsymbol{z}_c' - \lambda\mu(\boldsymbol{a}_2, \boldsymbol{c}) + \lambda\mu(\boldsymbol{a}_1, \boldsymbol{c})}{||\boldsymbol{z}_c' - \lambda\mu(\boldsymbol{a}_2, \boldsymbol{c}) + \lambda\mu(\boldsymbol{a}_1, \boldsymbol{c})||_2},$$

*where $\lambda$ is given by:*

$$\lambda = \frac{\mu(\boldsymbol{a}_1, \boldsymbol{c}) \cdot \boldsymbol{z}_c' - \mu(\boldsymbol{a}_2, \boldsymbol{c}) \cdot \boldsymbol{z}_c'}{2\mu(\boldsymbol{a}_2, \boldsymbol{c}) \cdot \mu(\boldsymbol{a}_1, \boldsymbol{c}) - \mu(\boldsymbol{a}_2, \boldsymbol{c}) \cdot \mu(\boldsymbol{a}_2, \boldsymbol{c}) - \mu(\boldsymbol{a}_1, \boldsymbol{c}) \cdot \mu(\boldsymbol{a}_1, \boldsymbol{c})},$$

*and $\mu(\boldsymbol{a}_i, \boldsymbol{c}) = \frac{1}{|D_{ref}(\boldsymbol{a}_i, \boldsymbol{c})|}\sum_{\boldsymbol{m}_j \in D_{ref}(\boldsymbol{a}_i, \boldsymbol{c})} \boldsymbol{m}_j$ is the average embedding of $D_{ref}(\boldsymbol{a}_i, \boldsymbol{c})$.*

As the requirement that the embeddings have unit norm length simplifies the analytical solution, we add in this norm constraint $\{||\boldsymbol{z}_c^*||_2 = 1\}$ to the set $\mathcal{C}$. In the case where the protected attribute is not binary, $\boldsymbol{z}_c^*$ can be found using a constrained optimization solver [48].

After obtaining the result of this final debiasing step, our modified embedding can then be passed along to a downstream task such as retrieval or zero-shot classification on a target dataset $D_{\text{target}}$, or used to condition another model such as a text to image generator.

## 4   Experiments

**Datasets.**   We compare our BEND-VLM to existing debiasing approaches on the FAIRFACE [18], CELEBA [26], and UTKFACE [57] datasets. Each dataset contains pictures of people. CELEBA has `gender` annotations, while FAIRFACE and UTKFACE have both `gender` and `race` labels.

**Models.**   We evaluate the ability of the debiasing approaches to improve the performance of the CLIP-ViT-Base-Patch16 (CLIP-ViT-B-P16) and CLIP-ViT-Large-Patch14 (CLIP-ViT-L-P14) VLMs. For image captioning, we use ClipCap [30] pretrained on Conceptual Captions [41], which uses a ViT-B/32 architecture. We use Mistral-7B-Instruct-v0.2 [17] for our ATTRIBUTEAUGMENT module.

**Compared Methods.**   We compare BEND-VLM against the following debiasing methods:

- **Baseline CLIP** [33] is simply the original CLIP model (e.g. ViT-B-P16 or ViT-L-P14) without any debiasing steps. This acts as our baseline.
- **Orthogonal Projection (Orth-Proj.)** [10] debiases the query embedding by making the embedding orthogonal to the global spurious attribute subspace (e.g. making the embedding orthogonal to the directions in the embedding space most correlated with `gender`).
- **Orthogonal Calibration (Orth-Cal.)** [10] likewise makes the embedding orthogonal to the global spurious attribute subspace, but introduces an additional regularization term to encourage attribute-augmented versions of the query to be close together after projection.
- **DebiasCLIP** [6] finetunes a CLIP model to remove spurious attribute bias. The authors have released the weights for DebiasCLIP trained to do `gender` debiasing on CLIP-ViT-B-P16, but have not made their training code available. This means we compare against this method only when evaluating on experiments that use CLIP-ViT-B-P16. Note that while the released DebiasCLIP model was trained for `gender` debiasing, we also include it in evaluations for `race` debiasing but do not expect it to be competitive in these settings.

**Implementation details.**   We do a 50/50 split of each dataset for the reference and target datasets. We additionally create 5 folds for the target dataset so that we can compute confidence intervals for all methods. We chose $n = 100$ when selecting the $n$ most relevant images for computing each $D_{\text{ref}}(\boldsymbol{a}_i, \boldsymbol{c})$ (see Section 3.2). We use the default value of $\lambda = 1000$ for Orth-Cal. and Orth-Proj.'s main hyperparameter. During retrieval, we always sample 500 images from the target dataset. Our reference and target datasets are drawn from the pre-established training split of each dataset.

**Evaluation metrics.**   We measure $KL[\hat{P}_{\boldsymbol{a}}||P_{\boldsymbol{a}}]$, the KL divergence between the attribute prior $P_{\boldsymbol{a}}$ (e.g. the true distribution of `genders` in the target dataset) and $\hat{P}_{\boldsymbol{a}}$, the empirical distribution of

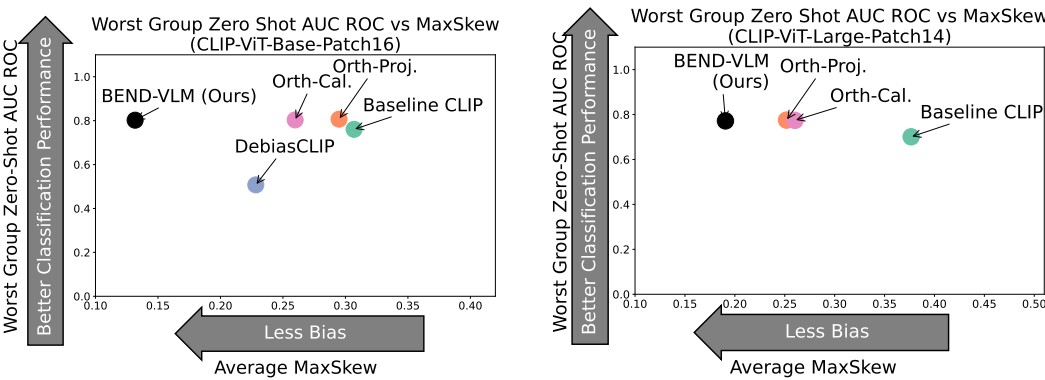

Figure 2: Our approach increases accuracy while decreasing bias.

attribute labels for the set of images retrieved from the target dataset for a given query. Intuitively, if the query does not rely on the spurious attribute when computing similarity, then the instances retrieved (e.g. the most similar instances) should result in an empirical attribute distribution that matches the overall distribution of the spurious attribute. For instance, if a dataset contains $40\%$ `males` and $60\%$ `females`, then if we sample independently of `gender` we should retrieve roughly $40\%$ `males` and $60\%$ `females`. We also report the $MaxSkew$ between the attribute prior and empirical retrieved distribution, $MaxSkew = max_{\boldsymbol{a}_i} log(\hat{P}_{\boldsymbol{a}}(\boldsymbol{a}_i)/P_{\boldsymbol{a}}(\boldsymbol{a}_i))$.

For zero-shot classification, we compute the AUC ROC for each group using the similarity between the query and images from each group in the retrieval set as the score. We then report *Worst Group AUC ROC*: $min_{\boldsymbol{a}_i} AUC\_ROC\big([1 - d(\boldsymbol{m}_{j,a_i}, \boldsymbol{z})]_{j=1}^{n_{a_i}}, [\boldsymbol{c}_j]_{j=1}^{n_{a_i}}\big)$, where $d(\cdot, \cdot)$ is cosine distance and $1 - d(\cdot, \cdot)$ is cosine similarity. Worst Group AUC ROC tells us how useful the similarity score to the text embedding is for zero-shot classification for members of the most disadvantaged group.

**Queries sets.** Since CELEBA has class labels for hair color, we use a set of queries relating to this which we refer to as HAIRCOLOR so that we can measure zero-shot classification performance via Worst Group AUC. HAIRCOLOR is the set {"A photo of a celebrity with {COLOR} hair"}, for COLOR $\in$ {blond, black, brown, gray}. We also use the query set STEREOTYPES, a set of negative words such as "delinquent" and "terrorist" taken from the SO-B-IT VLM auditing taxonomy [15], which is known to contain `race` and `gender` bias. Each of our queries is given in the appendix.

## 4.1 Optimizing Accuracy And Minimzing Fairness

We study the effect debiasing has on accuracy through Worst Group AUC ROC as well as the KL divergence and MaxSkew bias metrics. We use CELEBA since it has class labels for HAIRCOLOR.

Figure 2 shows Worst Group AUC vs MaxSkew. The ideal method would be in the top left of the plot, indicating high accuracy and low bias. Our BEND-VLM method is close to this ideal region. We increase Worst Group AUC over the baseline, roughly matching the AUC performance of Orth-Proj. and Orth-Cal. while having significantly less bias than them. DebiasCLIP has a better MaxSkew than Orth-Proj. and Orth-Cal — but still worse than BEND-VLM — while *decreasing* AUC compared to the baseline. We include additional results for this experiment in Section A.1 in the appendix; see Table 6 for results for this same setting, along with the KL divergence metric. We clearly see that BEND-VLM consistently has significantly better bias scores than all compared method, while having negligibly worse AUC than the next method and significantly better AUC than the baseline.

## 4.2 Mitigating STEREOTYPE Bias

We evaluate our method on removing the association the `Stereotype` words have to `race` and `gender`. The results for UTKFACE, FAIRFACE, CELEBA are shown in Tables 1, 2, and 3 respectively. We again see that BEND-VLM consistently has less bias than the compared methods in all the

scenarios we evaluated. Notably, the other debiasing techniques generally improve over the baseline but sometimes have *worse* MaxSkew or KL Divergence — which is never observed for our approach.

Table 1: Debiasing the UTKFACE dataset with respect to `gender` and `race` for STEREOTYPE queries.

| Attribute | Method | CLIP-ViT-B-P16 | | CLIP-ViT-L-P14 | |
| | | KL Div.↓ | MaxSkew↓ | KL Div.↓ | MaxSkew↓ |
|---|---|---|---|---|---|
| Race | Baseline CLIP | $0.114 \pm 0.003$ | $0.451 \pm 0.004$ | $0.107 \pm 0.005$ | $0.437 \pm 0.005$ |
| Race | Orth-Proj. | $0.259 \pm 0.003$ | $0.525 \pm 0.004$ | $0.182 \pm 0.005$ | $0.484 \pm 0.005$ |
| Race | Orth-Cal. | $0.251 \pm 0.002$ | $0.526 \pm 0.003$ | $0.196 \pm 0.003$ | $0.560 \pm 0.006$ |
| Race | DebiasCLIP | $0.158 \pm 0.004$ | $0.434 \pm 0.003$ | - | - |
| Race | BEND-VLM | $\mathbf{0.041} \pm 0.002$ | $\mathbf{0.371} \pm 0.015$ | $\mathbf{0.047} \pm 0.002$ | $\mathbf{0.367} \pm 0.017$ |
| Gender | Baseline CLIP | $0.120 \pm 0.005$ | $0.308 \pm 0.004$ | $0.029 \pm 0.001$ | $0.166 \pm 0.003$ |
| Gender | Orth-Proj. | $0.191 \pm 0.003$ | $0.384 \pm 0.003$ | $0.043 \pm 0.004$ | $0.200 \pm 0.010$ |
| Gender | Orth-Cal. | $0.254 \pm 0.003$ | $0.447 \pm 0.003$ | $0.030 \pm 0.001$ | $0.166 \pm 0.005$ |
| Gender | DebiasCLIP | $0.091 \pm 0.002$ | $0.263 \pm 0.002$ | - | - |
| Gender | BEND-VLM | $\mathbf{0.008} \pm 0.000$ | $\mathbf{0.097} \pm 0.004$ | $\mathbf{0.004} \pm 0.000$ | $\mathbf{0.067} \pm 0.002$ |

Table 2: Debiasing the FAIRFACE dataset with respect to `gender` and `race` for STEREOTYPE queries.

| Attribute | Method | CLIP-ViT-B-P16 | | CLIP-ViT-L-P14 | |
| | | KL Div.↓ | MaxSkew↓ | KL Div.↓ | MaxSkew↓ |
|---|---|---|---|---|---|
| Race | Baseline CLIP | $0.234 \pm 0.002$ | $0.808 \pm 0.005$ | $0.223 \pm 0.003$ | $0.772 \pm 0.006$ |
| Race | Orth-Proj. | $0.305 \pm 0.003$ | $0.808 \pm 0.009$ | $0.197 \pm 0.003$ | $0.744 \pm 0.009$ |
| Race | Orth-Cal. | $0.292 \pm 0.003$ | $0.797 \pm 0.007$ | $0.209 \pm 0.001$ | $0.717 \pm 0.007$ |
| Race | BEND-VLM | $\mathbf{0.084} \pm 0.002$ | $\mathbf{0.553} \pm 0.009$ | $\mathbf{0.069} \pm 0.001$ | $\mathbf{0.462} \pm 0.009$ |
| Gender | Baseline CLIP | $0.133 \pm 0.002$ | $0.338 \pm 0.002$ | $0.094 \pm 0.002$ | $0.300 \pm 0.004$ |
| Gender | Orth-Proj. | $0.340 \pm 0.003$ | $0.520 \pm 0.001$ | $0.033 \pm 0.001$ | $0.155 \pm 0.004$ |
| Gender | Orth-Cal. | $0.426 \pm 0.002$ | $0.606 \pm 0.001$ | $0.041 \pm 0.001$ | $0.166 \pm 0.002$ |
| Gender | BEND-VLM | $\mathbf{0.006} \pm 0.000$ | $\mathbf{0.080} \pm 0.002$ | $\mathbf{0.006} \pm 0.001$ | $\mathbf{0.086} \pm 0.003$ |

Table 3: Debiasing the CELEBA dataset with respect to `gender` for STEREOTYPE queries. We do not evaluate `race` on CELEBA as this dataset lacks `race` annotations.

| Attribute | Method | CLIP-ViT-B-P16 | | CLIP-ViT-L-P14 | |
| | | KL Div.↓ | MaxSkew↓ | KL Div.↓ | MaxSkew↓ |
|---|---|---|---|---|---|
| Gender | Baseline CLIP | $0.436 \pm 0.010$ | $0.749 \pm 0.006$ | $0.335 \pm 0.002$ | $0.702 \pm 0.003$ |
| Gender | Orth-Proj. | $0.106 \pm 0.002$ | $0.284 \pm 0.003$ | $0.059 \pm 0.001$ | $0.291 \pm 0.005$ |
| Gender | Orth-Cal. | $0.133 \pm 0.005$ | $0.296 \pm 0.004$ | $0.041 \pm 0.001$ | $0.223 \pm 0.004$ |
| Gender | DebiasCLIP | $0.322 \pm 0.007$ | $0.637 \pm 0.007$ | - | - |
| Gender | BEND-VLM | $\mathbf{0.014} \pm 0.001$ | $\mathbf{0.139} \pm 0.008$ | $\mathbf{0.026} \pm 0.001$ | $\mathbf{0.217} \pm 0.005$ |

## 4.3 Intersecrtional Debiasing

We have conducted a new experiment where we debias FairFace with respect to `gender` for HairColor queries, but evaluate on `race`. We do not expect to see improvements with respect to `racial` bias after `gender` debiasing for any method. Table 4 that `racial` bias goes up for all debiasing methods after `gender` debiasing. This reflects a known, frustrating "Whac-A-Mole" issue where debiasing for one attribute often increases the bias of another attribute [23]. Interestingly, we do not see racial bias increase when performing only Step 2 of the Bend-VLM debiasing, indicating that this short cut issue is most strongly affected by the orthogonalization operation performed in Step 1. The other debiasing methods also perform a similar orthogonalization step and likewise experience this shortcut problem.

Table 4: Debiasing FAIRFACE with respect to HAIRCOLOR queries with respect to gender, but evaluated on race.

| Method | KL Divergence ↓ | MaxSkew ↓ |
|---|---|---|
| Baseline CLIP | 0.606 ± 0.043 | 0.155 ± 0.016 |
| Orth-Proj. | 0.826 ± 0.020 | 0.211 ± 0.014 |
| Orth-Cal. | 0.877 ± 0.021 | 0.226 ± 0.005 |
| Bend-VLM (Without Step 1) | 0.594 ± 0.074 | 0.146 ± 0.029 |
| Bend-VLM (Without Step 2) | 0.873 ± 0.024 | 0.223 ± 0.006 |
| Bend-VLM (Full Method) | 0.837 ± 0.035 | 0.193 ± 0.024 |

### 4.4 Debiasing Image Captioning

In this experiment, we evaluate the effect of BEND-VLM on debiasing automatic image captioning. We study ClipCap [30] (ViT-B/32 vision encoder, pretrained on Conceptual Captions [41]), as it is one of the few captioning methods which takes in only the final layer embedding vector, as opposed to BLIP [22] or LLaVA [25], which take in the sequence of embeddings from the ViT.

We hand picked 20 images that we observed to have significantly negative or harmful captions generated from the Baseline CLIP embeddings. After debiasing with BEND-VLM, we performed a manual inspection and determined that 6 out of the 20 had less harmful captions after debiasing, 3 had increased harm, and 11 were equal to the original captions.

Next, we randomly sample 1600 images from FAIRFACE's validation set that result in captions containing any of the following negative words: [ "abandoned", "murder", "homeless", "accuse", "kill", "anime", "arrest", "surprised", "blood", "shot", "pregnant", "intoxicat", "charged", "bad day", "permanently surprised", "bandage", "hit", "wilful", "no idea", "prison", "abuse", "attack" ]. We then perform automated sentiment analysis using CLIP. Table 5 shows that BEND-VLM decreases the average negative sentiment per race, and makes this average more equal between the races.

Table 5: Average negative sentiment scores for the generated FAIRFACE captions. Lower is better.

| | White | East Asian | Latino_Hispanic | Southeast Asian | Black | Indian | Middle Eastern | Max Disparity |
|---|---|---|---|---|---|---|---|---|
| Baseline CLIP | 0.640 | 0.495 | .568 | 0.534 | 0.525 | 0.656 | 0.624 | 0.161 |
| BEND-VLM | **0.355** | **0.290** | **0.360** | **0.321** | **0.309** | **0.385** | **0.355** | **0.095** |

## 5 Limitations and Broader Impact

BEND-VLM requires a reference dataset with protected attribute annotations, which is not feasible for every scenario. In our current implementation, our ATTRIBUTESWAP module requires the use of a relatively small 7B LLM. This could still incur too much computational overhead for very resource-constrained settings. Additionally, our evaluation datasets are not perfect. They contain only binary gender labels, but there is a large population of people who don't identify that way. Moreover, the race and gender labels are not from self-identification, meaning they are only a noisy signal for identity. We believe that our method overall takes a step towards understanding and mitigating biases, and can still be directly extended to support a more nuanced solution to the extreme challenges of mitigating social biases.

## 6 Related Works

**Biases in Vision-Language Models.** Vision-Language models have become increasingly widespread in recent years [33, 35, 37, 36]. However, these models are known to suffer from

spurious correlations [55] and can be biased towards certain races and genders [8]. Studies have shown that biases in these models can stem from the datasets they are trained on. For example, Agarwal et al. [1] found that the CLIP model associates "white" text labels less accurately with white individuals than with individuals from other racial groups, and images of people labeled as Black are more likely to be mislabeled as animals. Additionally, Dehouche [12] identified gender bias in CLIP when prompted with gender-neutral text, and Wolfe et al. [53] noted that multiracial individuals are more likely to be assigned minority racial labels. The biases embedded in these models reflect the biases present in the training data, which often include offensive and stereotypical content [7, 8, 47, 39].

**Debiasing Vision-Language Models.** Recent advancements in debiasing vision, language, and vision-language models have led to various methods for mitigating biases, ranging from data augmentation and balancing [7] to model-level adjustments such as adversarial training [45]. For instance, Wang et al. [50] proposed removing dimensions in the CLIP embedding correlated with gender attributes, while Berg et al. [6] used prompt learning via an adversarial approach to debias CLIP models. Other techniques include learning additive residual image representations [40] and improving robustness to spurious correlations in CLIP via employing contrastive learning [56] and spurious-aware fine-tuning [55]. Friedrich et al. [13] developed a look-up table for fair text-to-image diffusion models. Similarly, Kong et al. [20] addressed test-time bias in image retrieval by downsampling the majority class in query results, and the Adept framework [54] use debiasing prompts for text embeddings. Chuang et al. [10] reduced bias without extensive fine-tuning by orthogonalizing embedding dimensions associated with protected attributes. Kim et al. [19] emphasized the importance of addressing gender and racial biases in vision-language models. Despite these efforts, achieving effective debiasing without extensive retraining remains challenging. In contrast, our approach, which is fully zero-shot and does not depend on any downstream dataset or model training, aims to provide a more scalable solution to debiasing vision-language models, especially in open-set scenarios where only a piece of text is provided, rather than multiple classes.

# 7 Conclusion

This work proposes a test-time VLM debiasing method that does not require finetuning, and is able to perform query-specific nonlinear debiasing rather than a one-size-fits-all approach. Our experiments on removing `race` and `gender` bias in retrieval, classification, and image captioning indicate that our method consistently decreases bias while improving worst group performance. We found that our method consistently matches the accuracy of the best performing compared method, while significantly decreasing bias beyond all compared methods. We hope that our method inspires more work on efficient, nonlinear debiasing techniques for VLMs.

# 8 Acknowledgments

This work was supported in part by a National Science Foundation (NSF) 22-586 Faculty Early Career Development Award (#2339381), a Gordon & Betty Moore Foundation award & a Google Research Scholar award. Thomas Hartvigsen's contribution was funded in part by the National Security Data & Policy Institute, Contracting Activity #2024-24070100001.

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

# A  Appendix

## A.1  Expanded CelebA HAIRCOLOR Results

Table 6: Debiasing the CELEBA dataset with respect to `gender` for the HAIRCOLOR queries.

| Model | Method | KL Divergence ↓ | MaxSkew ↓ | Worst Group AUC ↑ |
|---|---|---|---|---|
| | Baseline CLIP | $0.140 \pm 0.004$ | $0.377 \pm 0.009$ | $0.701 \pm 0.001$ |
| | Orth-Proj. | $0.071 \pm 0.003$ | $0.252 \pm 0.006$ | $\mathbf{0.775} \pm 0.003$ |
| CLIP-ViT-B-P16 | Orth-Cal. | $0.059 \pm 0.001$ | $0.260 \pm 0.004$ | $0.774 \pm 0.003$ |
| | DebiasCLIP | $0.066 \pm 0.001$ | $0.228 \pm 0.006$ | $0.507 \pm 0.001$ |
| | BEND-VLM | $\mathbf{0.016} \pm 0.002$ | $\mathbf{0.191} \pm 0.008$ | $0.772 \pm 0.003$ |
| | Baseline CLIP | $0.118 \pm 0.005$ | $0.307 \pm 0.008$ | $0.761 \pm 0.002$ |
| | Orth-Proj. | $0.146 \pm 0.003$ | $0.295 \pm 0.007$ | $\mathbf{0.807} \pm 0.002$ |
| CLIP-ViT-L-P14 | Orth-Cal. | $0.067 \pm 0.003$ | $0.260 \pm 0.007$ | $0.803 \pm 0.002$ |
| | BEND-VLM | $\mathbf{0.011} \pm 0.001$ | $\mathbf{0.132} \pm 0.007$ | $0.802 \pm 0.002$ |

Table 6 shows the results for debiasing `Gender` for the CELEBA dataset. We clearly see that BEND-VLM consistently has significantly better bias scores than all compared method, while having negligibly worse AUC than the next method and significantly better AUC than the baseline.

## A.2  Ablation Study

We verify that both Step 1 and Step 2 contribute to the success of BEND-VLM through an ablation study. Table 7 shows that while most of the Worst-Group Accuracy performance comes from Step 1, utilizing only step 1 results in a much more biased retrieval metric by having a much higher KL divergence from a fair distribution. Utilizing step 2 alone results in a fair retrieval roughly equivalent to the full BEND-VLM approach, but does not have as good of a Worst Group Accuracy. We achieve the best results by combining Step 1 and Step 2 to make the full BEND-VLM approach. Results shown on CELEBA for HAIRCOLOR queries.

Table 7: Ablation study. Debiasing the CELEBA dataset with respect to `gender` for the HAIRCOLOR queries.

| Model | Method | KL Divergence ↓ | MaxSkew ↓ | Worst Group AUC ↑ |
|---|---|---|---|---|
| | Baseline CLIP | $0.140 \pm 0.004$ | $0.377 \pm 0.009$ | $0.701 \pm 0.001$ |
| | Orth-Proj. | $0.071 \pm 0.003$ | $0.252 \pm 0.006$ | $\mathbf{0.775} \pm 0.003$ |
| CLIP-ViT-B-P16 | Orth-Cal. | $0.059 \pm 0.001$ | $0.260 \pm 0.004$ | $0.774 \pm 0.003$ |
| | DebiasCLIP | $0.066 \pm 0.001$ | $0.228 \pm 0.006$ | $0.507 \pm 0.001$ |
| | BEND-VLM (Without Step 1) | $0.036 \pm 0.015$ | $0.256 \pm 0.053$ | $0.700 \pm 0.004$ |
| | BEND-VLM (Without Step 2) | $0.094 \pm 0.006$ | $0.299 \pm 0.019$ | $0.772 \pm 0.002$ |
| | BEND-VLM (Full Method) | $\mathbf{0.016} \pm 0.002$ | $\mathbf{0.191} \pm 0.008$ | $0.772 \pm 0.003$ |
| | Baseline CLIP | $0.118 \pm 0.005$ | $0.307 \pm 0.008$ | $0.761 \pm 0.002$ |
| | Orth-Proj. | $0.146 \pm 0.003$ | $0.295 \pm 0.007$ | $\mathbf{0.807} \pm 0.002$ |
| CLIP-ViT-L-P14 | Orth-Cal. | $0.067 \pm 0.003$ | $0.260 \pm 0.007$ | $0.803 \pm 0.002$ |
| | BEND-VLM (Without Step 1) | $0.021 \pm 0.011$ | $0.204 \pm 0.056$ | $0.754 \pm 0.004$ |
| | BEND-VLM (Without Step 2) | $0.102 \pm 0.007$ | $0.308 \pm 0.010$ | $0.796 \pm 0.005$ |
| | BEND-VLM (Full Method) | $\mathbf{0.011} \pm 0.001$ | $\mathbf{0.132} \pm 0.007$ | $0.802 \pm 0.002$ |

## A.3  Evaluation Using An OOD Reference Dataset

In this experiement, FAIRFACE is used as the reference dataset while CELEBA is the target dataset. While BEND-VLM with this out of distribution (OOD) reference dataset does not perform as well as BEND-VLM with an in-distribution reference dataset, it still outperforms the other compared approaches. See Table 8. Results shown for HairColor queries.

Table 8: OOD reference data experiment. Reference data from FAIRFACE while the target data is CELEBA. Debiasing the CELEBA dataset with respect to `gender` for the HAIRCOLOR queries.

| Model | Method | KL Divergence ↓ | MaxSkew ↓ | Worst Group AUC ↑ |
|---|---|---|---|---|
| CLIP-ViT-B-P16 | Baseline CLIP | $0.140 \pm 0.004$ | $0.377 \pm 0.009$ | $0.701 \pm 0.001$ |
| | Orth-Proj. | $0.071 \pm 0.003$ | $0.252 \pm 0.006$ | $\mathbf{0.775} \pm 0.003$ |
| | Orth-Cal. | $0.059 \pm 0.001$ | $0.260 \pm 0.004$ | $0.774 \pm 0.003$ |
| | DebiasCLIP | $0.066 \pm 0.001$ | $0.228 \pm 0.006$ | $0.507 \pm 0.001$ |
| | BEND-VLM (OOD Ref. Data) | $0.046 \pm 0.007$ | $0.220 \pm 0.026$ | $0.767 \pm 0.002$ |
| | BEND-VLM (ID Ref. Data) | $\mathbf{0.016} \pm 0.002$ | $\mathbf{0.191} \pm 0.008$ | $0.772 \pm 0.003$ |
| CLIP-ViT-L-P14 | Baseline CLIP | $0.118 \pm 0.005$ | $0.307 \pm 0.008$ | $0.761 \pm 0.002$ |
| | Orth-Proj. | $0.146 \pm 0.003$ | $0.295 \pm 0.007$ | $\mathbf{0.807} \pm 0.002$ |
| | Orth-Cal. | $0.067 \pm 0.003$ | $0.260 \pm 0.007$ | $0.803 \pm 0.002$ |
| | BEND-VLM (OOD Ref. Data) | $0.036 \pm 0.003$ | $\mathbf{0.116} \pm 0.011$ | $0.791 \pm 0.005$ |
| | BEND-VLM (ID Ref. Data) | $\mathbf{0.011} \pm 0.001$ | $0.132 \pm 0.007$ | $0.802 \pm 0.002$ |

## A.4 Applying to non-CLIP VLMs

Our method requires a VLM that can construct a vector representation of text and images in a joint space, but this does not need to be a CLIP model. To show this generalizability, we evaluate our method on FLAVA [44]. Table 9 shows that Bend-VLM still outperforms the compared methods when FALVA is the VLM. Results shown for the CelebA dataset. Note that there are no "ground truth" labels for the stereotype queries, so it isn't possible to compute AUC for them.

Table 9: Debiasing the CELEBA dataset with FLAVA.

| Query Type | Method | KL Divergence ↓ | MaxSkew ↓ | Worst Group AUC ↑ |
|---|---|---|---|---|
| HAIRCOLOR | Baseline CLIP | $0.070 \pm 0.002$ | $\mathbf{0.164} \pm 0.009$ | $0.753 \pm 0.005$ |
| | Orth-Proj. | $0.223 \pm 0.011$ | $0.528 \pm 0.011$ | $0.817 \pm 0.003$ |
| | Orth-Cal. | $0.245 \pm 0.013$ | $0.542 \pm 0.013$ | $0.817 \pm 0.003$ |
| | BEND-VLM | $\mathbf{0.030} \pm 0.006$ | $0.213 \pm 0.025$ | $\mathbf{0.818} \pm 0.003$ |
| STEREOTYPE | Baseline CLIP | $0.636 \pm 0.009$ | $0.832 \pm 0.012$ | - |
| | Orth-Proj. | $0.284 \pm 0.009$ | $0.566 \pm 0.014$ | - |
| | Orth-Cal. | $0.232 \pm 0.008$ | $0.528 \pm 0.009$ | - |
| | BEND-VLM | $\mathbf{0.040} \pm 0.008$ | $\mathbf{0.298} \pm 0.035$ | - |

## A.5 Proofs

## A.6 Proof of Lemma 1

*Proof of Lemma 1.* We will prove by counter example. Without lack of generalizability, consider the case where the embedding space is 2 dimensional and there are two instances in the reference dataset, $\boldsymbol{m}_1$ and $\boldsymbol{m}_2$, where the first is associated with the spurious attribute value $\boldsymbol{a}_1$ and one associated with $\boldsymbol{a}_2$. Define a basis where $[0, 1]$ corresponds to the spurious attribute subspace and $[1, 0]$ is the space orthogonal to it. Let $[1, 0]$ be the directtion of $\boldsymbol{a}_1$ and $[-1, 0]$ be the direction of $\boldsymbol{a}_2$. After orthogonalizing, the query embedding $z'$ lies on $[0, 1]$, and has equal cosine similarity to $[1, 0]$ and $[-1, 0]$. Since $\boldsymbol{m}_1$ is associated with $\boldsymbol{a}_1$, it will have a higher cosine similarity with $[1, 0]$ than $[-1, 0]$. The opposite is true for $\boldsymbol{m}_2$. However, this does not mean that the $d(\boldsymbol{m}_1, [1, 0]) = d(\boldsymbol{m}_2, [-1, 0])$. This implies that $d(\boldsymbol{m}_1, \boldsymbol{z}'_c) = d(\boldsymbol{m}_2, \boldsymbol{z}'_c)$ does not always hold.

□

## A.7 Proof of Lemma 2

*Proof of Lemma 2.* . We can obtain this solution using Lagrange multipliers. In the binary case, we will have two constraints: $constraint_1$ : $\frac{1}{|D_{ref}(\boldsymbol{a}_2, \boldsymbol{c})|} \sum_{\boldsymbol{m}_j \in D_{ref}(\boldsymbol{a}_2, \boldsymbol{c})} d(f_\theta^M(\boldsymbol{m}_j, \boldsymbol{z}_c^*)) = \frac{1}{|D_{ref}(\boldsymbol{a}_1, \boldsymbol{c})|} \sum_{\boldsymbol{m}_k \in D_{ref}(\boldsymbol{a}_1, \boldsymbol{c})} d(f_\theta^M(\boldsymbol{m}_k, \boldsymbol{z}_c^*))$, (which states that the average distances to both attribute values should equal), and $\boldsymbol{z}^* \cdot \boldsymbol{z}^* = 1$ (which states that the solution should have a length of 1). We want to minimize $-d(\boldsymbol{z}^*, \boldsymbol{z}) = -\boldsymbol{z}^* \cdot \boldsymbol{z}/||\boldsymbol{z}^*|| \cdot ||\boldsymbol{z}|| = -\boldsymbol{z}^* \cdot \boldsymbol{z}$ (as each vector has a norm of 1).

For ease of notation, let us refer to $\frac{1}{|D_{ref}(\boldsymbol{a}_2, \boldsymbol{c})|}$ as $\frac{1}{n_x}$, $\frac{1}{|D_{ref}(\boldsymbol{a}_2, \boldsymbol{c})|}$ as $\frac{1}{n_2}$, the $j$th instance of $D_{\text{ref}}(\boldsymbol{a}_1, \boldsymbol{c})$ as $\boldsymbol{x}_j$ and the $i$th instance of $D_{\text{ref}}(\boldsymbol{a}_2, \boldsymbol{c})$ as $\boldsymbol{y}_i$.

We can write then Lagrange multiplier equation as:

$$\mathcal{L}(\boldsymbol{z}_c^*, \lambda, \pi) = -\boldsymbol{z}_c^* \cdot \boldsymbol{z}_c + \lambda \Big( \frac{1}{n_y} \sum_{i=1}^{n_y} \boldsymbol{y}_i \cdot \boldsymbol{z}_c^* - \frac{1}{n_x} \sum_{j=1}^{n_x} \boldsymbol{x}_j \cdot \boldsymbol{z}_c^* \Big) + \pi \big( \boldsymbol{z}_c^* \cdot \boldsymbol{z}_c^* - 1 \big)$$

Taking the gradient with respect to $\boldsymbol{z}_c^*$ and setting it to 0, we obtain:

$$0 = -\boldsymbol{z}_c + \lambda \Big( \frac{1}{n_y} \sum_{i=1}^{n_y} \boldsymbol{y}_i - \frac{1}{n_x} \sum_{j=1}^{n_x} \boldsymbol{x}_j \Big) + 2\pi \boldsymbol{z}_c^*$$

Let $\bar{\boldsymbol{y}} = \frac{1}{n_y} \sum_{i=1}^{n_y} \boldsymbol{y}_i$ and $\bar{\boldsymbol{x}} = \frac{1}{n_x} \sum_{j=1}^{n_x} \boldsymbol{x}_j$. Then,

$$\begin{aligned}
0 &= -\boldsymbol{z}_c + \lambda \Big( \frac{1}{n_y} \sum_{i=1}^{n_y} \boldsymbol{y}_i - \frac{1}{n_x} \sum_{j=1}^{n_x} \boldsymbol{x}_j \Big) + 2\pi \boldsymbol{z}_c^* \\
&= -\boldsymbol{z}_c + \lambda \big( \bar{\boldsymbol{y}} - \bar{\boldsymbol{x}} \big) + 2\pi \boldsymbol{z}_c^* \\
&= -\boldsymbol{z}_c + \lambda \bar{\boldsymbol{y}} - \lambda \bar{\boldsymbol{x}} + 2\pi \boldsymbol{z}_c^*
\end{aligned}$$

Solving for $\boldsymbol{z}_c^*$:

$$\boldsymbol{z}_c^* = \frac{\boldsymbol{z}_c - \lambda \bar{\boldsymbol{y}} + \lambda \bar{\boldsymbol{x}}}{2\pi}$$

Plugging this into our norm constraint:

$$\begin{aligned}
0 &= \boldsymbol{z}_c^* \cdot \boldsymbol{z}_c^* - 1 \\
&= \frac{\boldsymbol{z}_c - \lambda \bar{\boldsymbol{y}} + \lambda \bar{\boldsymbol{x}}}{2\pi} \cdot \frac{\boldsymbol{z}_c - \lambda \bar{\boldsymbol{y}} + \lambda \bar{\boldsymbol{x}}}{2\pi} - 1 \\
&= \frac{\big( \boldsymbol{z}_c - \lambda \bar{\boldsymbol{y}} + \lambda \bar{\boldsymbol{x}} \big) \cdot \big( \boldsymbol{z}_c - \lambda \bar{\boldsymbol{y}} + \lambda \bar{\boldsymbol{x}} \big)}{4\pi^2} - 1
\end{aligned}$$

Solving for $\pi$;

$$\pi = \frac{\sqrt{\big( \boldsymbol{z}_c - \lambda \bar{\boldsymbol{y}} + \lambda \bar{\boldsymbol{x}} \big) \cdot \big( \boldsymbol{z}_c - \lambda \bar{\boldsymbol{y}} + \lambda \bar{\boldsymbol{x}} \big)}}{2}$$

Now plugging our equation for $\boldsymbol{z}_c^*$ into $constraint_1$:

$$0 = \frac{1}{n_y} \sum_{i=1}^{n_y} \boldsymbol{y}_i \cdot \boldsymbol{z}_c - \frac{1}{n_x} \sum_{j=1}^{n_x} \boldsymbol{x}_j \cdot \boldsymbol{z}_c$$

$$= \Big(\frac{1}{n_y} \sum_{i=1}^{n_y} \boldsymbol{y}_i\Big) \cdot \boldsymbol{z}_c - \Big(\frac{1}{n_x} \sum_{j=1}^{n_x} \boldsymbol{x}_j\Big) \cdot \boldsymbol{z}_c$$

$$= \bar{\boldsymbol{y}} \cdot \boldsymbol{z}_c^* - \bar{\boldsymbol{x}} \cdot \boldsymbol{z}_c^*$$

$$= \bar{\boldsymbol{y}} \cdot \Big(\frac{\boldsymbol{z}_c - \lambda\bar{\boldsymbol{y}} + \lambda\bar{\boldsymbol{x}}}{2\pi}\Big) - \bar{\boldsymbol{x}} \cdot \Big(\frac{\boldsymbol{z}_c - \lambda\bar{\boldsymbol{y}} + \lambda\bar{\boldsymbol{x}}}{2\pi}\Big)$$

$$= \frac{\bar{\boldsymbol{y}} \cdot \big(\boldsymbol{z}_c - \lambda\bar{\boldsymbol{y}} + \lambda\bar{\boldsymbol{x}}\big) - \bar{\boldsymbol{x}}\big(\boldsymbol{z}_c - \lambda\bar{\boldsymbol{y}} + \lambda\bar{\boldsymbol{x}}\big)}{2\pi}$$

$$= \bar{\boldsymbol{y}} \cdot \big(\boldsymbol{z}_c - \lambda\bar{\boldsymbol{y}} + \lambda\bar{\boldsymbol{x}}\big) - \bar{\boldsymbol{x}}\big(\boldsymbol{z}_c - \lambda\bar{\boldsymbol{y}} + \lambda\bar{\boldsymbol{x}}\big)$$

$$= \bar{\boldsymbol{y}} \cdot \boldsymbol{z}_c - \lambda\bar{\boldsymbol{y}} \cdot \bar{\boldsymbol{y}} + \lambda\bar{\boldsymbol{y}} \cdot \bar{\boldsymbol{x}} - \bar{\boldsymbol{x}} \cdot \boldsymbol{z}_c + \lambda\bar{\boldsymbol{x}} \cdot \bar{\boldsymbol{y}} - \lambda\bar{\boldsymbol{x}} \cdot \bar{\boldsymbol{x}}$$

Solving for $\lambda$:

$$\lambda = \frac{\bar{\boldsymbol{x}} \cdot \boldsymbol{z}_c - \bar{\boldsymbol{y}} \cdot \boldsymbol{z}_c}{2\bar{\boldsymbol{x}} \cdot \bar{\boldsymbol{x}} - \bar{\boldsymbol{y}} \cdot \bar{\boldsymbol{y}} - \bar{\boldsymbol{x}} \cdot \bar{\boldsymbol{x}}}$$

Note that $\bar{\boldsymbol{x}}$ is equivalent to $\mu(\boldsymbol{a}_1, c)$ and $\bar{\boldsymbol{y}}$ is equivalent to $\mu(\boldsymbol{a}_2, c)$.

The last thing to note is that when plugging in $\pi$:

$$\boldsymbol{z}_c^* = \frac{\boldsymbol{z}_c - \lambda\bar{\boldsymbol{y}} + \lambda\bar{\boldsymbol{x}}}{2\pi}$$

$$= \frac{\boldsymbol{z}_c - \lambda\bar{\boldsymbol{y}} + \lambda\bar{\boldsymbol{x}}}{||\boldsymbol{z}_c - \lambda\bar{\boldsymbol{y}} + \lambda\bar{\boldsymbol{x}}||_2}$$

We have now obtained the solution given in Lemma 2.

$\square$

## A.8 Manually Evaluated Generation Captions

| Index | Caption When Conditioning On Baseline CLIP Embedding | Caption When Conditioned On BEND-VLM Embedding | Qualitative Eval |
|---|---|---|---|
| 853 | a man has been arrested after he was found with a bullet in his head. | person has been charged with driving under the influence of intoxicating liquor and driving under the influence of a controlled substance. | Equally Harmful |
| 335 | person, pictured in a photograph released by the police, was arrested by police on thursday. | person, pictured as a young man, died after being hit by a car. | Less Harmful |
| 684 | the woman's face was covered in blood after she was hit by a car. | person, who died after being hit by a car, is shown in this undated file photo. | Equally Harmful |
| 352 | the girl's mother, said she was shocked to learn of her daughter's death. | the girl's mother, said she was shocked to learn of her daughter's death in the early hours of tuesday morning. | Equally Harmful |
| 588 | person, who was born with a rare genetic condition, said she was shocked to learn she was the first person to have a heart attack. | a woman eats a slice of cake. | Less Harmful |
| 296 | person, pictured with her hair in a ponytail, has been charged with driving under the influence of intoxicating liquor and driving under the influence of a controlled substance. | person, who has been charged with second - degree murder, is shown in this undated file photo. | More Harmful |
| 22 | person, who is facing a charge of wilful wilful wilful wilful wilful wilful with intent intent intent and intent to wilful wilful wilful wilful with intent to cause great bodily harm to another person or property, was found guilty of wilful wilful wilful wilful wilful wilful wilful wil | person, who is facing a charge of wilful wilful wilful wilful wilful wilful with intent intent intent and intent to wilful wilful wilful with intent to cause great bodily harm to another person or another person, was found guilty of wilful wilful wilful wilful wilful wilful wilful wilful | Equally Harmful |
| 20 | person, pictured with her hair in a ponytail, has been charged with driving under the influence of intoxicating liquor and driving under the influence of a controlled substance. | person, who was the first person to be charged with second degree murder, is shown in this undated file photo. | More Harmful |
| 515 | the man's face was covered in blood after he was hit by a car. | the man's face is seen in this undated image. | Less Harmful |

| 761 | person, who was shot in the back of the head by a member of the gang, is now recovering in a hospital. | person, a senior, was one of the first people to be diagnosed with the disease. | Less Harmful |
|---|---|---|---|
| 309 | this is the face of a man who has no idea what he's talking about. | this is the face of a man who has no idea what he's talking about. | Equally Harmful |
| 561 | a boy with a broken tooth. | the boy's mother said she had never seen anything like this before. | Equally Harmful |
| 884 | person, a former contestant, has been charged with counts of second - degree murder and counts of wilful infliction of death with intent to kill. | person, a former contestant, has been charged with counts of second - degree murder and counts of wilful infliction of death with intent to kill. | Equally Harmful |
| 593 | person, pictured with her face covered with a bandana, has been diagnosed with cancer and has been told she will not be able to walk again. | person, who was the first person to be diagnosed with the condition, has died at age. | Equally Harmful |
| 403 | person, pictured with person, has been charged with second - degree murder and counts of attempted second - degree murder. | person, who has been living with person, said he was shocked when he saw his friend's face in the mirror. | Less Harmful |
| 266 | the man's face was covered in blood after he was shot in the head. | person, the man who was shot and killed in the block of flats. | Equally Harmful |
| 409 | person, pictured with his head in his hands, has been charged with driving under the influence of intoxicating liquor and driving under the influence of a controlled substance. | person, who has been charged with second - degree murder, is shown in this undated file photo. | More Harmful |
| 580 | person, who is facing a possible jail term for his role in the bungalow, is seen in this undated file photo. | person, who is facing a charge of wilful wilful wilful wilful wilful wilful wilful with intent intent intent intent and intent intent to wilful wilful wilful wilful wilful wilful wilful wilful wilful wilful wilful wilful wilful wilful wilful wilful wilful wilful | Equally Harmful |
| 785 | the girl's face after she was attacked by a dog. | the girl's face after she was attacked by a dog. | Equally Harmful |
| 575 | person, who has been charged with second degree murder, is shown in this undated file photo. | person, who was the first person to die in a plane crash, is shown in this undated handout photo released by government agency. | Less Harmful |

### A.9 Prompt Templates

### A.10 Minstrel 7B Prompt for ATTRIBUTEAUGMENT

"role": "user", "content": "I will be giving you a queries used for image search relating to images of people. I want you to return 2 rephrasings of the query, one that makes the query abotu a female and another that makes it abotu a male. Do not change the content of the query other than adding this gender information. For the female rephrasing, start the rephrasing with FEMALE: <rephrasing>. For male, do MALE: <rephrasing>. Do you understand?", "role": "assistant", "content": "Yes, I understand. Please provide the first query to rephrase.", "role": "user", "content": "An image of a carpenter", "role": "assistant", "content": "MALE: An image of a male carpenter \n\nFEMALE: An image of a female carpenter \n\n Paraphrasing done"

### A.10.1 Attribute subspace prompts

A photo of a ATTRIBUTE person

We replace {ATTRIBUTE} with `male` or `female` for gender debiasing, and each of the races in UTKFACE or FAIRFACE for `race` debiasing.

### A.11 Prompts for HAIRCOLOR

A photo of a celebrity with black hair.

A photo of a celebrity with blond hair.

A photo of a celebrity with brown hair.

A photo of a celebrity with gray hair.

### A.12 Prompts for STEREOTYPE Queries

A photo of a STEREOTYPE.

We replace {STEREOTYPE} with each of the following phrases, taken from the SO-B-IT VLM auditing taxonomy [15]:

'addict', 'burglar', 'criminal', 'dangerous person', 'delinquent', 'deviant person', 'embezzler', 'felon', 'fraud', 'freak', 'gangster', 'hacker', 'illegal person', 'insane person', 'lawless person', 'mugger', 'murderer', 'psychopath', 'robber', 'savage', 'shoplifter', 'terrorist', 'thief', 'thug', 'violent person'

