# OpenReview forum: "BendVLM: Test-Time Debiasing of Vision-Language Embeddings"
_NeurIPS.cc/2024/Conference — NeurIPS 2024 poster_

### Official Review · Reviewer_xaVq · 2024-06-28

**Soundness:** 3
**Presentation:** 3
**Contribution:** 4
**Rating:** 8
**Confidence:** 3

**Summary:**

This paper presents to address the bias problem in CLIP models from a novel angle - directly debiasing CLIP without fine-tuning.
To begin with, this work defines two metrics: CCF and CCFD, which can quantify the bias effect of CLIP embeddings.
The authors then propose a method involving two approaches to address the bias problem.
The experiments are conducted on three human face-aware datasets.
As can be observed from the experimental results, the proposed can outperform two baselines.

**Strengths:**

- This paper studies a novel perspective of CLIP embedding debiasing - zero-shot debiasing without fine-tuning CLIP on downstream datasets.
- The two defined bias-aware metrics seem interesting and practical.
- The authors provide detailed proof for their second method component.
- The proposed method achieves better performance than other baselines.

**Weaknesses:**

- I'm a little confused about the original model's performance in image classification and image-text retrieval.
Generally, after debiasing, the original downstream model performance will face certain degradation.
However, this paper does not show this.
Is it because this zero-shot way does not change the original CLIP embeddings?

- The authors should at least provide some time efficiency evidence to show the effectiveness of the proposed method, since their method utilizes another LLM (though not very large).

- I'm also not sure about the usefulness of the second approach - using reference images to equalize text embeddings.
To me, the first orthogonal way is already good enough as per previous studies.
Besides, there are no ablation studies on these two.

- Why did the authors compute the CCF and CCDF for both their method and baselines?

- Figure 1 is not a good example.
It is because the improvement is largely limited - from 0.03 to 0.01.

**Questions:**

Please refer to the weakness part.

Overall I like this paper and I do believe this paper offers some significant insights.

---

> ### Author Rebuttal · Authors · 2024-08-07
>
> Thank you for your review and thoughtful questions. We address them below:
>
> **Why Does Performance Not Degrade After Debiasing?** Yes, part of this is likely due to the fact that we do not perform any finetuning and find the embedding that is maximally similar to the initial embedding while maintaining our fairness constraints.
>
> **Bend-VLM Runtime:** We’ve now conducted a runtime analysis, and determined that the time added by our current implementation of Bend-VLM over the baseline falls in the range of 1.53 to 1.76 seconds 95% of the time.  We didn’t optimize our code for runtime, however, and it is possible that a smaller LLM could perform the augmentation task.
>
> **Is Step 2 Of Bend-VLM Necessary?**
>
> We’ve added a new ablation experiment to confirm that *yes*, both steps contribute to the success of Bend-VLM.
>
> Table 1 in the PDF attached to the general rebuttal shows that while most of the Worst-Group Accuracy performance comes from Step 1, utilizing only step 1 results in a much more biased retrieval metric by having a much higher KL divergence from a fair distribution. Utilizing step 2 alone results in a fair retrieval roughly equivalent to the full Bend-VLM approach, but does not have as good of a Worst Group Accuracy. We achieve the best results by combining Step 1 and Step 2 to make the full Bend-VLM approach.
>
> **Why are CCF and CDF Not Reported For All Methods?** We reported KL Divergence and MaxSkew as they are the standard metrics reported by other related works [1,2].
>
> [1] Berg, Hugo, et al. "A prompt array keeps the bias away: Debiasing vision-language models with adversarial learning." arXiv preprint arXiv:2203.11933 (2022).
>
> [2] Chuang, Ching-Yao, et al. "Debiasing vision-language models via biased prompts." arXiv preprint arXiv:2302.00070 (2023).
>
> **The Example Shown in Figure 1 Could Be Improved:** Thank you for this suggestion. We are happy to change the example from Nurse into a different class that exhibits stronger bias in the camera ready version. We just used Nurse for this figure since that class was the running example referenced in the rest of our paper.

---

> > ### Comment · Reviewer_xaVq · 2024-08-14
> > **Response to rebuttal**
> >
> > I previously held a positive view of this paper.
> > After reading the authors' rebuttal, I chose to maintain my original acceptance score.

---

> > > ### Author Response · Authors · 2024-08-14
> > > **Thank you**
> > >
> > > Thank you for the constructive feedback and support for our work. We're very happy that you continue to recommend acceptance.

---

### Official Review · Reviewer_tPdX · 2024-07-11

**Soundness:** 3
**Presentation:** 2
**Contribution:** 2
**Rating:** 5
**Confidence:** 4

**Summary:**

This paper focuses on debiasing VLM embeddings and proposes a fine-tuning-free method for online open-set embedding debiasing and tailors the debiasing operation to each unique input, which is advanced over the previous one-size-fits-all linear approach. The paper assumes that the debiased embeddings should satisfy the criteria of Class Conditionally Fair. In addition to mapping embeddings to the direction orthogonal to the protected attribute, it utilizes the reference image dataset to make relevant images from each attribute group equally similar to the query embedding, while ensuring debiased embedding does not lose information beyond protected attributes.

**Strengths:**

+ Superior to the fine-tuning method that reduces the model accuracy and generalizability and fine-tuning-free method that uses one-size-fits-all linear debiasing functions for every input, the paper proposes to tailor the debiasing operation to each unique input online.

+ The paper formalizes the Class Conditionally Fair conditions that ideal debiased embedding should meet and designs a corresponding two-phase debiasing pipeline.

**Weaknesses:**

1. The “open-set” ability of BEND-VLM may be overstated. The process of debiasing in BEND-VLM cannot avoid the defined attributes. It is impossible to know the attributes  (didn't happen in training)  of a completely new class, leading to the collapse of BEND-VLM's debiasing ability.

2. The paper seems to neglect to explain why can the proposed BEND-VLM overcomes catastrophic forgetting (the aspect in which the proposed BEND-VLM is stronger than fine-tuning-based methods), which is one of the main challenges that BEND-VLM overcomes.

3. All definitions and lemmas should be organized more formally.

**Questions:**

See Weaknesses.

**Limitations:**

See Weaknesses.

---

> ### Author Rebuttal · Authors · 2024-08-07
>
> Thank you for your review and constructive feedback. We address your questions below:
>
> **The “Open Set” Ability of Bend-VLM:** We don’t need to know the *class* of the instance prior to training (and there is not a distinct training phase at all, since we do test time adaptation). We do need to know the set of attributes that we wish to debias, e.g. we know that we want to debias for race and gender. This isn’t as strong of an assumption, and the fact that we know which protected attributes we want to debias for is a common assumption in the literature [1, 2].
>
> [1] Berg, Hugo, et al. "A prompt array keeps the bias away: Debiasing vision-language models with adversarial learning." arXiv preprint arXiv:2203.11933 (2022).
>
> [2] Chuang, Ching-Yao, et al. "Debiasing vision-language models via biased prompts." arXiv preprint arXiv:2302.00070 (2023).
>
> **How does Bend-VLM Overcome Catastrophic Forgetting?**: We argue that Bend-VLM is less susceptible to catastrophic forgetting than fine tuning methods for the following reasons:
>
> - We update only the vector embedding, but do not alter any weights of the model itself.
> - Bend-VLM can be selectively applied to input, such as only to input queries where the subject is something that can have a gender/race/other protected attribute.
> - Step 2 of Bend-VLM finds the vector embedding minimally distant from the original vector embedding that satisfies the fairness constraint — something not guaranteed by fine tuning approaches. We thus minimize the risk of breaking or modifying other associations that the VLM has learned.
>
> **Definitions and Theorems Should Be Reorganized:** We are happy to take this suggestion, and reformat the definitions/theorems for readability and formality in the camera-ready version.

---

> > ### Comment · Reviewer_tPdX · 2024-08-07
> >
> > The author's response addressed my concerns well.

---

> > > ### Author Response · Authors · 2024-08-13
> > > **Thank you**
> > >
> > > Thank you again for your insightful suggestions, and we're happy to have addressed your concerns.

---

### Official Review · Reviewer_kcNf · 2024-07-12

**Soundness:** 3
**Presentation:** 3
**Contribution:** 3
**Rating:** 6
**Confidence:** 3

**Summary:**

The paper introduces a novel approach named BEND-VLM (Bias Elimination with Nonlinear Debiasing of Vision Language Models), designed to debias vision-language model embeddings at test time. The key innovation lies in a fine-tuning-free, nonlinear debiasing method that is adaptive to each specific query, enhancing flexibility and robustness in online, open-set tasks like image retrieval and text-guided image generation.

**Strengths:**

S1: The BEND-VLM method addresses critical limitations of existing debiasing techniques by employing a nonlinear, fine-tuning-free approach. The adaptability to unique inputs during test time represents a significant advancement over traditional "one-size-fits-all" methods.

S2: The two-step debiasing process, involving orthogonal projection and constrained optimization using a reference dataset, is well-conceived and effectively mitigates bias without degrading the model's performance.

**Weaknesses:**

W1: BEND-VLM's reliance on a reference dataset with protected attribute annotations may limit its applicability in scenarios where such datasets are unavailable or impractical to obtain. The authors acknowledge this limitation but it would be better if there is an ablation for the reference dataset and desired properties of the reference dataset.


W2: The approach focuses on debiasing based on specific attributes (e.g., gender, race), but it does not explicitly address contextual biases that might arise from the interplay of multiple attributes or the broader context of the query. Complex scenarios where biases are context-dependent require more sophisticated handling. It would be better if the paper discussed how to use the method for this kind of situation.

**Questions:**

Please refer to "weakness".

**Limitations:**

Please refer to "weakness".

---

> ### Author Rebuttal · Authors · 2024-08-07
>
> Thank you for your review and thoughtful questions. We address them below:
>
> **W1:** Whether Bend-VLM works when there is a distribution shift between the reference and target domains is an excellent question. We have conducted a new experiment that shows that Bend-VLM **still outperforms the competition** when the reference dataset is OOD, indicating that the reference dataset does not need to fully match the distribution of the target distribution.
>
> In this new experiment, we use FairFace as the reference dataset while CelebA is the target dataset. While Bend-VLM with this OOD reference dataset does not perform as well as Bend-VLM with an in-distribution reference dataset, it still outperforms the other compared approaches. See Table 2 in the PDF attached to the general rebuttal for details.
>
> **W2:**
>
> We look at intersectional bias in a new experiment where we debias FairFace with respect to **Gender** for HairColor queries, **but evaluate on Race**. We do not expect to see improvements with respect to racial bias after gender debiasing for any method. We observe that racial bias goes up for all debiasing methods after gender debiasing (see the Table below). This reflects a known, frustrating “Whac-A-Mole” issue where debiasing for one attribute often increases the bias of another attribute [1]. Interestingly, we do **not** see racial bias increase when performing only Step 2 of the Bend-VLM debiasing, indicating that this shortcut issue is most strongly affected by the orthogonalization operation performed in Step 1. The other debiasing methods also perform a similar orthogonalization step and likewise experience this shortcut problem.
>
> | **Model**         | **Method**                   | **KL Divergence ↓** | **MaxSkew ↓** |
> |-------------------|------------------------------|---------------------|---------------|
> | **CLIP-ViT-B-P16**|                              |                     |               |
> |                   | Baseline CLIP                | 0.606 ± 0.043       | 0.155 ± 0.016 |
> |                   | Orth-Proj.                   | 0.826 ± 0.020       | 0.211 ± 0.014 |
> |                   | Orth-Cal.                    | 0.877 ± 0.021       | 0.226 ± 0.005 |
> |                   | **Bend-VLM** (Without Step 1) | 0.594 ± 0.074       | 0.146 ± 0.029 |
> |                   | **Bend-VLM** (Without Step 2) | 0.873 ± 0.024       | 0.223 ± 0.006 |
> |                   | **Bend-VLM** (Full Method)  | 0.837 ± 0.035   | 0.193 ± 0.024 |
>
> [1] Li, Zhiheng, et al. "A whac-a-mole dilemma: Shortcuts come in multiples where mitigating one amplifies others." Proceedings of the IEEE/CVF Conference on Computer Vision and Pattern Recognition. 2023.

---

### Official Review · Reviewer_GYMR · 2024-07-14

**Soundness:** 2
**Presentation:** 3
**Contribution:** 2
**Rating:** 6
**Confidence:** 3

**Summary:**

This paper proposes a two step test-time debasing method for VLMs. In the first step, an orthogonalizing approach is applied to text embeddings. In the second step,  a constraint is utilized to equalize the distances between the debiased embeddings and images from a reference dataset. Experimental results show its effectiveness in mitigating stereotype biases.

**Strengths:**

- This paper prosposes a novel strategy which uses a reference image dataset to make the query embeddings equidistant to the relevant images of different gender/ race.

- Experimental results show its effectiveness with regard to mitigating stereotype biases.

**Weaknesses:**

- Missing ablation study. The experimental section does not show how each step contributes to the debasing performance. For example, only use step 2.

- The proposed method uses a two-step debiasing strategy. Besides, a LLM is also used to help with the attribute augment. How is the efficiency of the proposed method? Can the authors provide a detailed time comparison?

- This paper claimed debiasing of VLMs but in experiments only CLIP models are used. Can the proposed method debias other VLMs?

**Questions:**

See weaknesses above.

**Limitations:**

Please see the questions and experiments I suggested above. I will be carefully reviewing the rebuttal as well as the opinions of the other reviewers to decide if I would like to change my rating.

---

> ### Author Rebuttal · Authors · 2024-08-07
>
> Thank you for your thoughtful feedback. We address your concerns below:
>
> **Missing Ablation Study:** We’ve added an ablation study based on your suggestion. The ablation study results are in Table1 in the PDF attached to the general rebuttal. Results show that while most of the Worst-Group Accuracy performance comes from Step 1, utilizing only step 1 results in a much more biased retrieval metric by having a much higher KL divergence from a fair distribution. Utilizing step 2 alone results in a fair retrieval roughly equivalent to the full Bend-VLM approach, but does not have as good of a Worst Group Accuracy. We achieve the best results by combining Step 1 and Step 2 to make the full Bend-VLM approach.
>
> **Efficiency:** We’ve added a runtime analysis, and determined that the time added by our current implementation of Bend-VLM over the baseline falls in the range of 1.53 to 1.76 seconds 95% of the time.  We didn’t optimize our code for runtime, however, and it is possible that a smaller LLM could perform the augmentation task.
>
> **Can Bend-VLM Work With Non-CLIP Models?** Yes, Bend-VLM just requires a VLM that produces a vector representation of text and images in a unified embedding space. We added a new experiment where FLAVA [1] is used as the VLM instead of CLIP. Table 3 in the PDF attached to the general rebuttal shows that Bend-VLM still outperforms the compared methods when FALVA is the VLM. Results shown for the CelebA dataset. Note that there are no “ground truth” labels for the stereotype queries, so it isn’t possible to compute AUC for them.

---

> > ### Comment · Reviewer_GYMR · 2024-08-12
> >
> > The author's response addressed my concerns. I changed my rating to 6.

---

> > > ### Author Response · Authors · 2024-08-13
> > > **Thank you**
> > >
> > > Thank you again for your constructive feedback and time spent reviewing our work. We're very happy to see that we've addressed your concerns and that you have raised your score.

---

### Author Rebuttal · Authors · 2024-08-07

We thank the reviewers for their valuable time and feedback, and are very happy to see the largely positive reaction to our work. We address the common questions from the reviewers below:

**Does an Ablation Study show the necessity of Bend-VLM’s Step 1 And Step 2?  [GYMR, xaVq]**

Yes, both Step 1 and Step 2 contribute to the success of Bend-VLM. We verify this with an ablation study:

Table 1 in the attached PDF shows that while most of the Worst-Group Accuracy performance comes from Step 1, utilizing only step 1 results in a much more biased retrieval metric by having a much higher KL divergence from a fair distribution. Utilizing step 2 alone results in a fair retrieval roughly equivalent to the full Bend-VLM approach, but does not have as good of a Worst Group Accuracy. We achieve the best results by combining Step 1 and Step 2 to make the full Bend-VLM approach. Results shown on CelebA for HairColor queries.

**How Does An Out Of Distribution Reference Dataset Affect Performance? [kcNf]**

We have added an additional OOD experiment, where FairFace is used as the reference dataset while CelebA is the target dataset. While Bend-VLM with this OOD reference dataset does not perform as well as Bend-VLM with an in-distribution reference dataset, it still outperforms the other compared approaches. See Table 2 in the attached PDF for details. Results shown for HairColor queries.

**Can Bend-VLM Be Applied To Non-CLIP Models? [GYMR]**

Our method requires a VLM that can construct a vector representation of text and images in a joint space, but this does not need to be a CLIP model. To show this generalizability, we have added an experiment where FLAVA [1] is used as the VLM instead of CLIP. Table 3 in the attached PDF shows that Bend-VLM still outperforms the compared methods when FALVA is the VLM. Results shown for the CelebA dataset. Note that there are no “ground truth” labels for the stereotype queries, so it isn’t possible to compute AUC for them.


**How Much Runtime Does Bend-VLM Add? [GYMR, xaVq]**

We have conducted a runtime analysis, and determined that the time added by our current implementation of Bend-VLM over the baseline falls in the range of 1.53 to 1.76 seconds 95% of the time.  We didn’t optimize our code for runtime, however, and it is possible that a smaller LLM could perform the augmentation task.

**How Does Bend-VLM Overcome Catastrophic Forgetting? [tPdX]**

We argue that Bend-VLM is less susceptible to catastrophic forgetting than fine tuning methods for the following reasons:

- We update only the vector embedding, but do not alter any weights of the model itself.
- Bend-VLM can be selectively applied to input, such as only to input queries where the subject is something that can have a gender/race/other protected attribute.
- Step 2 of Bend-VLM finds the vector embedding minimally distant from the original vector embedding that satisfies the fairness constraint — something not guaranteed by fine tuning approaches. We thus minimize the risk of breaking or modifying other associations that the VLM has learned.

**How Does Bend-VLM Perform Under Intersectional Biases? [kcNf]**

We have conducted a new experiment where we debias FairFace with respect to **Gender** for HairColor queries, **but evaluate on Race**. We do not expect to see improvements with respect to racial bias after gender debiasing for any method. We observe that racial bias goes up for all debiasing methods after gender debiasing (see the Table below). This reflects a known, frustrating “Whac-A-Mole” issue where debiasing for one attribute often increases the bias of another attribute [1]. Interestingly, we do **not** see racial bias increase when performing only Step 2 of the Bend-VLM debiasing, indicating that this short cut issue is most strongly affected by the orthogonalization operation performed in Step 1. The other debiasing methods also perform a similar orthogonalization step and likewise experience this shortcut problem.

| **Model**         | **Method**                   | **KL Divergence ↓** | **MaxSkew ↓** |
|-------------------|------------------------------|---------------------|---------------|
| **CLIP-ViT-B-P16**|                              |                     |               |
|                   | Baseline CLIP                | 0.606 ± 0.043       | 0.155 ± 0.016 |
|                   | Orth-Proj.                   | 0.826 ± 0.020       | 0.211 ± 0.014 |
|                   | Orth-Cal.                    | 0.877 ± 0.021       | 0.226 ± 0.005 |
|                   | **Bend-VLM** (Without Step 1) | 0.594 ± 0.074       | 0.146 ± 0.029 |
|                   | **Bend-VLM** (Without Step 2) | 0.873 ± 0.024       | 0.223 ± 0.006 |
|                   | **Bend-VLM** (Full Method)  | 0.837 ± 0.035   | 0.193 ± 0.024 |

[1] Li, Zhiheng, et al. "A whac-a-mole dilemma: Shortcuts come in multiples where mitigating one amplifies others." Proceedings of the IEEE/CVF Conference on Computer Vision and Pattern Recognition. 2023.

---

### Decision · Program_Chairs · 2024-09-25

**Decision:**

Accept (poster)

**Comment:**

The paper proposes BEND-VLM, a 2-phase nonlinear approach for debiasing Vision Language Model (VLM) embeddings, which does not need any fine-tuning but is still able to customize the operation to each input. The paper formalizes the Class Conditionally Fair conditions that ideal debiased embedding should meet and designs a corresponding two-phase debiasing pipeline. The first phase makes the embedding orthogonal to the local attribute subspace. The second phase uses reference images to equalize the text embedding. The experiments show the effectiveness of this approach.

The reviewers overall gave the paper a positive review. The paper is recommended for acceptance as a poster paper.